# Green Solvents in the Extraction of Bioactive Compounds from Dried Apple Cultivars

**DOI:** 10.3390/foods12040893

**Published:** 2023-02-19

**Authors:** Marcela Hollá, Veronika Pilařová, František Švec, Hana Sklenářová

**Affiliations:** Department of Analytical Chemistry, Faculty of Pharmacy in Hradec Králové, Charles University, 50005 Hradec Králové, Czech Republic

**Keywords:** apple cultivars, carbon dioxide, extraction approaches, greenness evaluation, polyphenols, gas-expanded liquid extraction, ultrasound extraction

## Abstract

New extraction protocols, gas-expanded liquid extraction (GXLE), and ultrasound extraction (UE) have been optimized with an emphasis on using green solvents and maximizing the extraction of 14 selected phenolic compounds, including flavonoid-based compounds and phenolic acids from dried apples. The design of the experiments’ approach was applied to optimize the main extraction parameters. Fine tuning included optimization of the flow rate in GXLE and the extraction time for GXLE and UE. Optimized GXLE was carried out with CO_2_–ethanol–water (34/53.8/12.2; *v*/*v*/*v*) at a flow rate of 3 mL/min at a temperature of 75 °C and pressure of 120 bar for 30 min. UE with ethanol–water 26/74 (*v*/*v*) lasted for 10 min at 70 °C. Both methods differed in solvent consumption and sample throughput, while providing a comparable total phenolic content of 2442 µg/g with an RSD < 10% and 2226 µg/g with RSD < 6%, for GXLE and UE, respectively. Both methods were used in determining the phenolic compounds in five apple cultivars, ‘Angold’, ‘Artiga’, ‘Golden Delicious’, ‘Meteor’, and ‘Topaz’. Phenolic profiles were plotted with chlorogenic acid, catechin, epicatechin, hirsutrin, phloridzin, and guaiaverin as the main components. Statistical evaluation, including pair t-test, Bland–Altman test, and linear regression did not reveal any differences between UE and GXLE results.

## 1. Introduction

Apples (fruits of *Malus domestica*) are one of the most important and worldwide cultivated fruit as they are a rich source of dietary phytonutrients. The apple is typically composed of 85% water, then mostly carbohydrates, including polysaccharides and sugars, and the last 1% representing bioactive compounds, such as vitamins, carotenoids, trace elements, fiber, phytosterols, and phenolics [1]. Apple phenolics can be divided into (i) nonflavonoid compounds such as gallic, caffeic, ferulic, and other phenolic acids, and (ii) flavonoid-based compounds, such as derivatives of flavonol, flavan-3-ol, anthocyanidin, and dihydrochalcone. Generally, phenolic compounds are considered polar to medium polar compounds that are prone to degradation after exposure to radiation, oxidation, and elevated temperature [2]. Thus, their extraction from plant material can be challenging due to the polar matter of the matrix and the presence of compounds with similar physicochemical properties.

Both conventional and green nonconventional extraction approaches have been applied for the extraction of phenolic compounds. Simple and robust maceration and Soxhlet extraction belong among the main conventional approaches still commonly used in many laboratories despite their several drawbacks, including long extraction time, solvent consumption, and possible degradation of the analytes of interest [3]. During the last few years, nonconventional approaches enabling the application of environmentally friendly solvents, reduction in extraction time, decrease in solvent consumption, and low energy consumption, have been preferred. These included microwave-assisted extraction, ultrasound extraction (UE), and supercritical fluid extraction (SFE) in different formats, including gas-expanded liquid extraction (GXLE) and pressurized liquid extraction (PLE) [3].

UE is a simple extraction approach that is often used for the extraction of polar compounds from fruit matrices. Its optimization is straightforward and features sufficient extraction efficiency. Typically, a polar solvent is selected with respect to the compound solubility, while frequency, temperature, and time must be optimized. Moreover, new trends, such as the application of deep eutectic solvents, enhancement of extraction by pressure, electric field, enzymes, and application of supercritical fluids are also reported [4,5,6]. The UE was applied for the extraction of phenolic antioxidants from turkey berry fruits using ethanolic extractant [7], phenolic compounds from Opuntia flowers extracted by ethanol [8], pectin and phenolics from apple peel by acetone [9], and phenolics from waste leaves extracted by aqueous solvents [10]. Another application was focused on the preservation quality and extraction of fresh-cut kiwifruit with a methanolic–aqueous mixture with high-intensity ultrasound extraction [11]. Sonication potentials in the extraction of phytoconstituents included the optimization and application of the extraction of bioactive compounds (oils, pectin, and protein) as discussed in detail [12].

On the other hand, SFE is the extraction approach originally designed for the extraction of nonpolar compounds using supercritical carbon dioxide (sCO_2_). Considered a nonpolar solvent with unique physicochemical properties, sCO_2_ is widely discussed elsewhere [13]. To increase the polarity of sCO_2_, the pressure and temperature need to be tuned, or a polar cosolvent miscible with sCO_2_ has to be added from negligible amounts to high levels at which the supercritical state is not achieved, and the liquid is called gas-expanded [1,13,14,15,16,17]. The GXLE principle can be successfully applied for the extraction of polar compounds such as phenolics, and several reports focused on GXLE of flavonoids and other phenolic compounds, mainly using sCO_2_ and ethanol as a cosolvent, have been published. They included the use of GXLE in the extraction of garlic phenolic compounds with methanol as the cosolvent [18], silymarin compounds with an ethanolic cosolvent [19], triterpenoids from acacia leaves extracted by ethyl acetate as the cosolvent [20], quercetin from medicinal plants [21], and caffeine from green tea [22], where combinations of CO_2_ with other green solvents such as ethyl lactate and ethyl acetate were also applied. SFE/GXLE and its efficiency were also compared with PLE in a study focused on the extraction of sea buckthorn pomace where compounds with different polarities were obtained [23]. Naeem et al. optimized UE with SFE for the extraction of polyphenols from various fruits, namely banana, orange, apple, onion, and garlic husks [24]. They confirmed that the application of higher pressure in SFE/GXLE enabled the extraction of more phenolics compared to UE carried out with water, methanol, and ethanol as extraction solvents.

The aim of our study was to develop UE and extraction with carbon dioxide as a part of the extraction solvents using the design of experiment approach enabling multivariate optimization of selected parameters to assess the effect of individual parameters, as well as their possible interactions. UE was selected due to its simplicity and availability of the instrumentation, while SFE/GXLE were involved in our study due to the application of nontoxic solvents and an increasing number of applications to polar analytes. Despite the fact that both methods are commonly used in many application fields, we wanted to compare them, not only in the terms of the extracted yield of selected phenolic compounds, their repeatability, and specificity using the statistical data but also in the terms of greenness, including toxic solvent and energy consumption and waste production, the importance of which has been increasing in recent years. Moreover, new protocols were applied for the determination of the phenolics content in different apple cultivars in the form of dried matter and the applicability in this field was confirmed for both methods.

## 2. Materials and Methods

### 2.1. Chemicals and Reagents

Commercially available standards of phenolics mostly present in different apple cultivars, including gallic acid (97.5–102.5%), chlorogenic acid (≥95%), (-)-epicatechin (≥90%), catechin (≥98%), caffeic acid (≥98%), rutin hydrate (≥94%), quercetin (≥95%), guaiaverin (≥99.3%), quercitrin (≥97%), hyperoside (≥98.35%), hirsutrin (≥90.1%), reynoutrin (≥97%), phloridzin (≥99%), and phloretin (≥99%), were purchased from Sigma–Aldrich (Steinheim am Albuch, Germany). The LC/MS grade solvents acetonitrile (ACN), methanol (MeOH), and absolute ethanol (EtOH, ≥99.7%) were supplied by Sigma–Aldrich (Prague, Czech Republic). Formic acid (≥98%) was purchased from Honeywell (Seelze, Germany). Ultrapure CO_2_ (>99.99%) in cylinders with a dip tube was provided by Messer (Prague, Czech Republic). Ultrapure water was acquired using a Milli-Q purification system (Millipore, Billerica, MA, USA).

### 2.2. Preparation of the Standard Solutions

Stock standard solutions for each analyte were prepared by weighting and dissolution of each solid compound in MeOH to obtain a concentration of 1.0 mg/mL. The working standard solutions containing 0.2 mg/mL were individually prepared by dilution in methanol–water (60/40, *v*/*v*) containing 0.1% formic acid (diluent mixture) to preserve the analytes’ stability. A mixed standard solution was prepared from all 14 working standard solutions via dilution with the diluent mixture to obtain a concentration of 10 µg/mL. All solutions were stored in the dark at 4 °C and prepared fresh every month. A diluent mixture was used for the subsequent dilution of extracts and mixed standard solutions.

### 2.3. Dried Apple Samples

Dried apple slices, including both pulp and peel, obtained in local supermarkets were used together with samples of dried apple slices with pulp and peel of locally bred cultivars to broaden the spectrum of polyphenols. These cultivars, namely ‘Meteor’, ‘Golden Delicious’, ‘Topaz’, ‘Artiga’, and ‘Angold’, were obtained from the Research and Breeding Institute of Pomology Holovousy Ltd. They were cut into slices and immediately dried in a Steba ED fruit dryer with five drying plates and electronic temperature control. The first step of pre-drying lasted for about 1 h at 50 °C defined by the water content in the raw material. Then, the main drying step was accomplished at 60 °C for 7 h. The dried apple slices were ground using a powerful kitchen blender (Sencor Super Blender SBU 7730BK, Říčany, Czech Republic) to obtain a powder with homogenously distributed parts of apple peel and pulp. Consequently, the sieving of the powder was carried out using a manual set of sieves, and particles sized from 0.5 to 2.5 mm were used for the extraction. A stock of homogenized dried apple samples was stored in sealed containers in the dark at 4 °C and used for the optimization of extraction methods. Individual samples of the same cultivars were prepared using the identical protocol and extracted applying optimized extraction conditions.

### 2.4. Analysis of the Active Compounds Using UHPLC-DAD

UHPLC chromatographic system Acquity UPLC (Waters, Milford, MA, USA) comprising a binary pump, an autosampler, a column oven, and a diode array detector (DAD) was used for the determination of active compounds in the extracts. The system control, data acquisition, and data evaluation were executed using the Empower software (Waters, Milford, MA, USA). The reversed-phase Triart ExRS C18 (150 × 3 mm; 1.9 µm) column preceded by a guard column Ascentis Express C18 (5 × 4.6 mm) packed with 5 µm particles was used for the separations. The mobile phases consisted of aqueous acetic acid with pH 2.8 (A) and acetonitrile (B) with a flow rate of 0.35 mL/min. The gradient elution enabled the separation of the active compounds. The gradient started with 10% B in A as the initial conditions. Then, B was ramped to 22% in 8 min and 28% in the next 2.2 min. Then, a steep increase to 40% in 30 s followed by a further increase to 50% in 3 min and held for 0.2 min. Finally, the percentage of B was decreased to initial conditions in 0.2 min and held for 3.4 min to equilibrate the system. The temperature of the column was 30 °C. The sample was cooled at 6 °C in an autosampler and 2 µL were injected into the UHPLC system. The separation of all tested phenolic compounds lasted 15 min.

The analytes summarized in Appendix A were detected by DAD at 4 different wavelengths: (i) 254 nm selected for guaiaverin, hirsutrin, hyperoside, reynoutrin, quercitrin; (ii) 280 nm selected for gallic acid, epicatechin, catechin, phloridzin, phloretin; (iii) 320 nm applied for chlorogenic and caffeic acid; and (iv) 354 nm selected for rutin and quercetin. The 3D record was also recorded for wavelengths in the range of 210–400 nm to collect UV spectra for all detected peaks. Obtained UV spectra, together with retention times, were used for the identification of individual active compounds by comparing standard solutions and extracts.

Concentration levels for each analyte were calculated from the respective integrated peak areas compared to the standard (10 µg/mL) while within the framework of the previous measurements for the determination of phenolics in apples (unpublished results). Calibrations of all analytes proved to be linear in the whole tested range starting from 0.1 µg/mL.

### 2.5. Design of Experiments

Design of experiments (DoE) was applied to optimize extraction conditions for both, SFE and UE, using MODDE 12.1 (Umetrics, Umeå, Sweden) software. Box-Behnken and D-optimal designs were selected as they enable the study of the effects of individual parameters and their interactions on the extracted amounts of phenolics. The individual DoE are described below. For the evaluation of designs, a multilinear regression method was used to calculate the fitting model and response surface. The adequacy of the models was evaluated by the R^2^ and Q^2^ values (R^2^ represents the model fit and Q^2^ is the estimate of future prediction precision), model validity, and reproducibility. The optimum processing conditions for maximizing the extracted amount of targeted active compounds were obtained by using graphical and numerical analysis based on the criteria of the desirability function and the response surface plots.

### 2.6. Extraction Methods

#### 2.6.1. Extraction Using Carbon Dioxide, SFE, and GXLE

CO_2_-based extractions were carried out using the SFE system (Waters MV-10, Milford, MA, USA) comprising a fluid delivery module with a high-pressure CO_2_ pump and the pump for the cosolvent, an oven holding the extraction vessels, an automated back pressure regulator, and a fraction collector module. The heads of the CO_2_ pump were cooled using a chiller operated at 5 °C. The flow rate of the extraction solvent was controlled as a volumetric ratio between CO_2_ and the cosolvent and was kept at 2 mL/min. Dynamic extraction mode with a continuous flow rate was used. After each extraction, the system was flushed with a CO_2_–cosolvent mixture for 5 min followed by neat CO_2_ to remove residual cosolvent from the capillaries. The system was controlled by ChromScope^TM^ software (Waters, Milford, MA, USA). The dried and milled sample (0.5 g) was placed in a 5 mL stainless steel extraction vessel (21 mm inner length, 19.8 mm ID) between two layers of 3 mm glass beads. The volume of collected extract was measured, and 1 mL aliquot was evaporated using Concentrator plus 5305 (Eppendorf, Hamburg, Germany) at 45 °C and reconstituted in 100 µL of the diluent mixture by intense shaking for 10 min using a shaker TS-100 (Biosan, Prague, Czech Republic). The reconstituted extract was centrifuged at 4341× *g* at 4 °C, and the obtained supernatant was transferred to the total recovery vials and immediately analyzed.

Our method was optimized in two steps. First, DoE using the Box-Behnken design with three center points was applied to select the conditions for major parameters, including (i) CO_2_–cosolvent ratio in the range from 10 to 70% (*v*/*v*); (ii) water addition to EtOH as a cosolvent mixture in the range 5–20%; (iii) extraction pressure of 100–300 bar; and (iv) extraction temperature of 30–80 °C. The extraction time during DoE was 10 min.

Consequently, the extraction kinetic was evaluated to optimize the extraction time and solvent flow rate. In these experiments, flow rates of 2, 3, 4, and 5 mL/min were tested, while individual fractions at defined time periods were collected at 5, 10, 15, 30, 45, and 60 min during one extraction procedure to plot the extraction kinetic curves. Finally, the size of the glass beads with a diameter of 2, 3, and 5 mm, and their effect on the extraction procedure was evaluated.

In the optimized GXLE protocol, the sample was mixed with 2 mm glass beads, and extraction was carried out with CO_2_–EtOH + 18.5% H_2_O at a ratio of 34/66 (*v*/*v*) at a flow rate of 3 mL/min. Extraction pressure was set at 120 bar and the temperature was kept at 75 °C. The extraction was completed in 30 min.

#### 2.6.2. Ultrasound Extraction

UE was carried out using a digital sonication bath DU-32 (Argo Lab, Italy; ultrasonic power 120 W and volume 3.2 L) with temperature regulation and sonication frequency fixed at 40 kHz. The sample (0.5 g) was placed in the centrifuge tube and 10 mL of extraction solvent was added. Once the extraction was completed, each tube with the sample was centrifuged using MPW 260R (MPW, Warszawa, Poland) at 4341× *g* and 4 °C for 10 min. The supernatant was filtered through a 0.22 µm PTFE filter (Chromservis, Prague, Czech Republic), and collected. Aliquots were treated using the same procedure described in Section 2.6.1 and stored at −20 °C until analyzed.

The D-optimal design was selected in DoE to optimize 2 important parameters, namely solvent composition and extraction temperature. The volume of solvent used for extraction was selected from previous experiences where the solubility of the tested phenolic compounds was considered. EtOH with a water addition of 0–100% and temperature in the range of 30–70 °C were tested in 15 experiments with 4 center points. All experiments were carried out in duplicates and the average concentration was used for the data evaluation. Consequently, the extraction kinetic was tested to plot the kinetic curve within the same defined time periods. The final protocol used 10 mL of EtOH for extraction of 0.5 mg sample in 10 min.

## 3. Results and Discussion

Two different extraction approaches considered green were selected as suitable for the extraction of bioactive phenolic compounds from dried apples. All 14 compounds summarized in Appendix A, including flavonoids, their glycosidic forms, and phenolic acids, were selected based on our previous study and experience with major phenolic compounds present in the different cultivars of apples [25,26,27]. Moreover, phloretin was included in the analysis as a marker of phloridzin decomposition under inappropriate conditions such as high temperature or undesirable temperature changes. UHPLC-DAD detection provided sufficient selectivity and resolution for all analytes including isomeric pairs shown in the chromatogram (Figure 1). The low resolution of catechin and caffeic acid did not affect the real samples analysis, as caffeic acid was not present in the tested apple extracts.

### 3.1. Development of the Extraction Method Using CO_2_

The optimal conditions for the SFE/GXLE method were obtained using DoE in MODDE software. The ranges of tested parameters were set with an emphasis on the physicochemical properties of the selected monitored bioactive compounds, including their polarity and molecular weight. Most of the tested compounds are polar to medium polar substances, some of them with a sugar constituent. Thus, high solubility in neat nonpolar supercritical carbon dioxide was not expected. Therefore, the extraction solvent included polar ethanol as a cosolvent in the range of 30–90% (*v*/*v*) in CO_2_. Moreover, 5–20% water was added to EtOH to increase the solvent polarity and compound solubility. The extraction temperature ranging from 30 to 80 °C and extraction pressure in the range of 100–300 bar were tested in accordance with instrument limitations and the possible risk of the thermal degradation of analytes at higher temperatures. The single-phase composition of suggested solvents mixtures composed of carbon dioxide, EtOH, and water in different ratios was checked by plotting the individual mixtures in a ternary phase diagram [28].

The extracted quantities of selected compounds were monitored as the total phenolic content (TPC) and are summarized in Table 1. TPC was determined as a sum of extracted amounts for all selected compounds. The extracts were dominated by eight analytes, including epicatechin, catechin, chlorogenic acid, phloridzin, guaiaverin, hirsutrin, reynoutrin, and quercitrin. Their content in the TPC exceeded 90%. The main DoE model parameters are summarized in Appendix A. The replicates plot shows the difference in TPC among the tested conditions (green points) and confirms good repeatability via three extractions carried out under the same conditions (blue squares). Summary of the fit model with R^2^ 0.87, Q^2^ 0.76, and their difference, <0.30, confirmed good model linearity and predictability of the future results. A validity value of 0.61 and reproducibility of 0.95 confirmed that the variation of the replicates was sufficient. The box plot summarizing the effects of individual parameters and their interactions confirmed that CO_2_ content with water addition was the most important factor affecting the obtained TPC while extraction temperature and pressure had a negligible effect. Indeed, an increase in cosolvent amount and higher water content in the cosolvent increased extracted TPC as expected, due to the increased polarity of the solvent. Moreover, two significant interactions among the parameters were confirmed. The normal distribution of residuals was also observed (Appendix A). Considering the valid model, it is obvious from the contour plot in Appendix A that extracted amounts of selected compounds increased with increasing water content in cosolvent, extraction temperature, and pressure. In contrast, increasing CO_2_ content in the extraction solvent decreased the extraction recovery as the polarity of the extraction solvent was also reduced. Twenty optimal conditions, O1–O20, summarized in Appendix A, were suggested by the MODDE Optimizer with an emphasis on maximizing the TPC extraction, with CO_2_ varying in a range of 10–52% and water content always over 8.9%. As the temperature and pressure did not have any significant effect on extracted amounts of phenolic compounds, their settings varied in the range of 35–80 °C and 120–300 bar, respectively.

The optimal setup suggested by MODDE (experiment O15, Appendix A) had the highest probability of failure, 28%. Therefore, two other experiments, O and R (Appendix A) were generated, including robust and optimal set points with a probability of failure < 1%. All 22 experiments were carried out and TCP was determined with the highest extraction amount in a range of 1477–1717 µg/g for experiments O13, O14, and O20 where the probability of failure was <6.1%. To avoid the failure of the extraction protocol, the repeatability of these experiments was verified with three replicates and expressed as a relative standard deviation (RSD). O13 enabled the extraction of 1650 µg/g TPC with 3.7% RSD, O14 extracted 1411 µg/g with 4.8% RSD, and the TCF content in O20 was 1586 µg/g, with 7.0% RSD. Based on these experiments, the conditions for O13, namely an extraction solvent composed of 34/53.8/12.2 CO_2_–EtOH–water mixture (*v*/*v*/*v*), an extraction temperature of 75 °C, and a pressure of 120 bar were finally selected as the optimum. The single-phase composition of the solvent was verified by plotting as a ternary diagram [28] and after that, the conditions were used in the following optimization steps.

The optimization of the flow rate and extraction time (Appendix A) revealed that a change in the flow rate from 2 mL/min to higher flow rates, especially at the beginning of the extraction, increased the extracted quantities of the selected compounds. This phenomenon resulted from the solubility–partitioning limitations of the extraction where compounds are better extracted in higher solvent volumes. The extracted amounts were, after 30 min, comparable at all the tested flow rates, 2, 3, 4, and 5 mL/min, and the extraction plateau was achieved. Based on the extraction solvent composition, and the comparable results for flow rates 3, 4, and 5 mL/min, the flow rate of 3 mL/min in 30 min was selected for the final GXLE method.

The last step of the method optimization focused on the evaluation of the size of the glass beads that were used as a dispersant of the sample. Indeed, their size, as well as the size of the sample, can affect the amount of extracted phenolics due to the possible heterogeneous flow velocities through the packed sample. This is the well-known channeling effect [29]. Thus, three different bead sizes, 2, 3, and 5 mm, were used while applying the optimal conditions. While decreasing the size, the total extracted amount of phenolics increased and the individual RSD decreased (Appendix A). With smaller beads and an increase in their surface area, the contact of the sample with solvent was better and a higher amount was extracted. Thus, 2 mm glass beads were used. Finally, the intraday and interday repeatability under the optimized conditions were evaluated and expressed as RSD 4.8 and 10.8%, respectively.

### 3.2. Development of Ultrasound Extraction

The conditions for UE were optimized via D-optimal DoE suggested by MODDE software. Although the ultrasound frequency is the main parameter that needs to be optimized in UE to achieve the highest amounts of extracted target compounds, it was not possible to optimize it in our study since the frequency in the used instrument was fixed. Thus, only the extraction solvent composition and temperature could be examined in the 15 experiments shown in Table 2. Ethanol, water, and their respective mixtures were selected as the polar solvents enabling sufficient solubility of phenolic compounds, while the elevated temperatures in the range from 30 to 70 °C were selected with respect to instrumentation limits, stability of the sample, and ethanol boiling point (78 °C). Similar to GXLE, TPC was monitored here with the same eight dominant compounds in produced extracts. The DoE model parameters are summarized in Appendix A. The plot for replicates confirmed perfect method repeatability under two tested conditions shown as blue squares, while green points show the difference in extracted TPC. Linearity (R^2^), validity (Q^2^), and reproducibility of the model were >0.99 as proved in the summary of fits, with a model validity of 0.35 which was considered sufficient. Both selected parameters had a significant effect on the extracted amount and one interaction was also indicated (coefficient plot, Appendix A). Indeed, the lower EtOH content and the higher extraction temperature enabled an increase in the extracted TPC which was also confirmed in the contour plot shown in Appendix A. Ten experiments were suggested by the MODDE optimizer with a 0% probability of failure, as shown in Appendix A. It is obvious that the proposed conditions for all experiments differed only in tenths of a percent of EtOH content, while results using a temperature of 60 °C differed significantly compared to conditions where 70 °C were proposed.

Thus, only two experiments in triplicates were carried out to confirm the final conditions for UE. The determined TPCs for both methods were comparable, in the range from 2167 to 2244 µg/g with an RSD < 1.8%. Therefore, UE carried out at 70 °C with 26% EtOH as an extraction solvent was selected for further optimization. In the next step, the kinetic of extraction was evaluated with respect to the extraction time. The results summarized in Appendix A show a significant increase in extracted TPC after 10 min and no further increase in the extracted amount was observed. However, even a slight decrease in extracted amount was observed after 30 min. This could result from a long exposure of the sample to the ultrasound and high temperature followed by the compound degradation and/or its bonding back to the plant matrix. Nevertheless, the decrease was still negligible and within the error of the method. Finally, 10 min extraction time was selected as optimum, and interday and intraday repeatability of the method was verified with 3 replicates as 5.8 and 4.9% RSD, respectively.

### 3.3. Comparison of Optimized Approaches and Application to Apple Cultivars

Both optimized methods aimed at the maximized quantity of the extracted phenolic compounds. GXLE under optimized conditions extracted 2442 µg/g TPC with 4.8% and 10.8% intraday (n = 3) and interday (n = 9) RSD, respectively. Extracted TPC was slightly lower for UE, 2226 µg/g TPC with 5.8% and 4.9% intraday (n = 3) and interday (n = 9) RSD, respectively.

GXLE and UE were applied to five different apple cultivars. The TPC determined for individual cultivars summarized in Figure 2A showed the agreement between both methods. Indeed, these results were confirmed by simple linear regression, where the correlation for the phenolics concentrations obtained by GXLE and UE was observed with an R^2^ always > 0.97 (Figure 2B). The only significant difference between extraction methods was observed with the variety ‘Angold’. TPC from GXLE was 1227 µg/g (2.5% RSD) while it was 1738 µg/g with 0.3% RSD for UE. This discrepancy could be caused by the coextraction and coelution of structurally close compounds with similar physicochemical properties. Despite that, the overall difference between both extraction approaches was negligible. Phenolic profiles for each individual cultivar could then be created. According to Figure 2A, each cultivar is characterized by a different composition (Appendix A).

Chlorogenic acid, catechin, epicatechin, hirsutrin, guaiaverin, phloridzin, and hyperoside were the main components in all cultivars. For example, chlorogenic acid represented 59% TPC in ‘Angold’ and 41% in ‘Artiga’ while its amount in ‘Meteor’ (12%) was comparable with other major phenolics. Guaiaverin (25%) was the dominant phenolic compound in ‘Topaz’. Overall, plots for ‘Topaz’ and ‘Meteor’ featured similar profiles, however, with different TPC. Additionally, a higher content of catechin was monitored in the ‘Meteor’ and ‘Golden Delicious’ cultivars. Our results are in good agreement with reports published previously [26,27] where ‘Angold’ was characterized by a high content of chlorogenic acid while ‘Meteor’ and ‘Golden Delicious’ with catechin and epicatechin as the major compounds. Comparing TPC content, ‘Angold’ was characterized by quite high phenolics content, while ‘Topaz’ and ‘Golden Delicious’ were the cultivars with a medium content, and ‘Meteor’ revealed the lowest TPC. Nevertheless, it should be emphasized that comparison is always affected by growing conditions, humidity, sunshine, and other external conditions in the individual harvest.

The chromatographic profiles of extracts for all cultivars compared with standard solutions are summarized in Appendix A. It is obvious that GXLE provided slightly more selective extraction without any significant noise and ballast peaks compared to the UE counterparts. Since the individual compounds were determined in extracts obtained using both methods, the conformity of the methods was determined as a correlation using linear regression. This test summarized in Figure 3 for main quantified compounds, including catechin, chlorogenic acid, guaiaverin, hirsutrin, hyperoside, phloridzin, reynoutrine, and rutin confirmed a strong correlation between the results of both GXLE and UE with correlation coefficients R^2^ > 0.94. A significant difference was observed only for epicatechin, where a higher amount was monitored using UE (R^2^ = 0.28) caused by an outlier for epicatechin content in ‘Angold’. Caffeic acid, gallic acid, quercetin, and quercitrin were not quantified due to the detected peaks being below the lowest calibration point of 0.1 µg/mL. The same results were obtained using a paired t-test (Appendix A) and Bland–Altman test (Appendix A) with only two outliers found. These points were identified as epicatechin and chlorogenic acid extracted from the ‘Angold’ cultivar using UE thus confirming previous results.

The optimized extraction methods were compared with the previously published methods applied for fruit matrices and the extraction of bioactive substances including phenolics. The optimized conditions are in agreement with the previously published methods mainly using EtOH [3,4,5,6,7,8,19,22] as an extraction solvent considered green. EtOH extraction strength can be easily tuned by mixing with other solvents, including water, and ultrasound intensity can also help to increase the extraction effectivity [11].

Finally, the greenness of the extraction methods was evaluated using the free software Analytical GREEnness (AGREE, v. 05 beta) calculator based on the 12 green chemistry principles [30,31,32]. The main focus was kept on the sample preparation and extraction approach as the phenolic compounds were determined by the same UHPLC method. For this reason, the parameter weights were applied to emphasize the extraction part of the analysis. Most of the criteria applied for the greenness evaluation are the same as demonstrated in Figure 4. Both protocols involved external sample preparation or offline sample preparation (criterion #1) using only 0.5 g of dried apple tissue (#2). All experiments and measurements were carried out offline (#3) using several procedure steps such as sample weighting, extraction, evaporation, reconstitution, and UHPLC analysis (#4). Both methods do not require any derivatization step (#6). Moreover, some of the used solvents are biobased, nontoxic, and nonflammable (#10, #11, #12). The main differences between UE and GXLE were associated with the semiautomation or manual sample handling (#5), waste production (#7), energy consumption (#9), and the number of samples extracted per hour (#8). Indeed, to extract the same quantities of phenolics, the GXLE protocol required larger volumes of solvents including CO_2_, EtOH, and water, while UE was carried out only with 10 mL of a solvent mixture composed of EtOH and water. Moreover, the GXLE extraction time was three times longer and only two samples were extracted per hour. On the other hand, UE required only 10 min, and the parallel extraction could be carried out. It would allow the extraction of six samples per hour even if it would be possible to place only one sample in the sonication bath.

The total scores of AGREE evaluation with emphasized weaknesses of both methods are summarized in Figure 4. It is obvious from the score plots that UE is greener with a score of 0.71 while GXLE had a score of 0.61. Detailed protocols are summarized in Appendix A.

## 4. Conclusions

We developed two extraction methods using environmentally-friendly solvents for the determination of phenolic compounds in dried apples. Both, GXLE and UE, were carefully optimized using DoE with an emphasis on maximizing the extracted amount of individually selected phenolics as well as TPC. The composition of extraction solvents was the crucial parameter. In GXLE, the ternary mixture of CO_2_–EtOH–water in a ratio of 34/53.8/12.2 (*v*/*v*/*v*) at 3 mL/min flow rate, a temperature of 75 °C, and a pressure of 120 bar was applied for 30 min to extract TPC. UE used a polar mixture containing EtOH–water in a ratio of 26/74 (*v*/*v*) and 70 °C was the best temperature. Due to the smaller consumption of solvent and energy, a higher throughput, and instrument availability, UE was found to be a ‘greener’ approach compared to the optimized GXLE. That, in contrast, was considered more suitable for common laboratory practice. However, both methods produced similar results in the terms of repeatability (RSD < 10%) and extracted amounts of selected phenolic compounds as confirmed via statistical data evaluation. Both methods were successfully applied in the extraction of five different apple cultivars with diverse phenolic profiles. It is likely that our methods can also be used for the extraction of phenolics from other apple varieties and perhaps even from other types of fruit.

## Figures and Tables

**Figure 1 foods-12-00893-f001:**
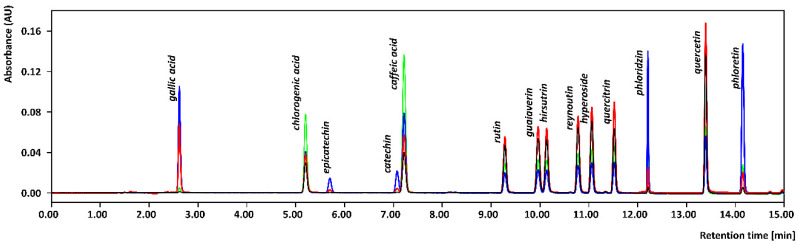
UHPLC-UV chromatogram of standards at a concentration of 10 µg/mL in MeOH–water + 0.1% formic acid (60/40, *v*/*v*) using optimized separation conditions and detected at 4 different wavelengths: 254 nm in red, 280 nm in blue, 320 nm in green, and 354 in black.

**Figure 2 foods-12-00893-f002:**
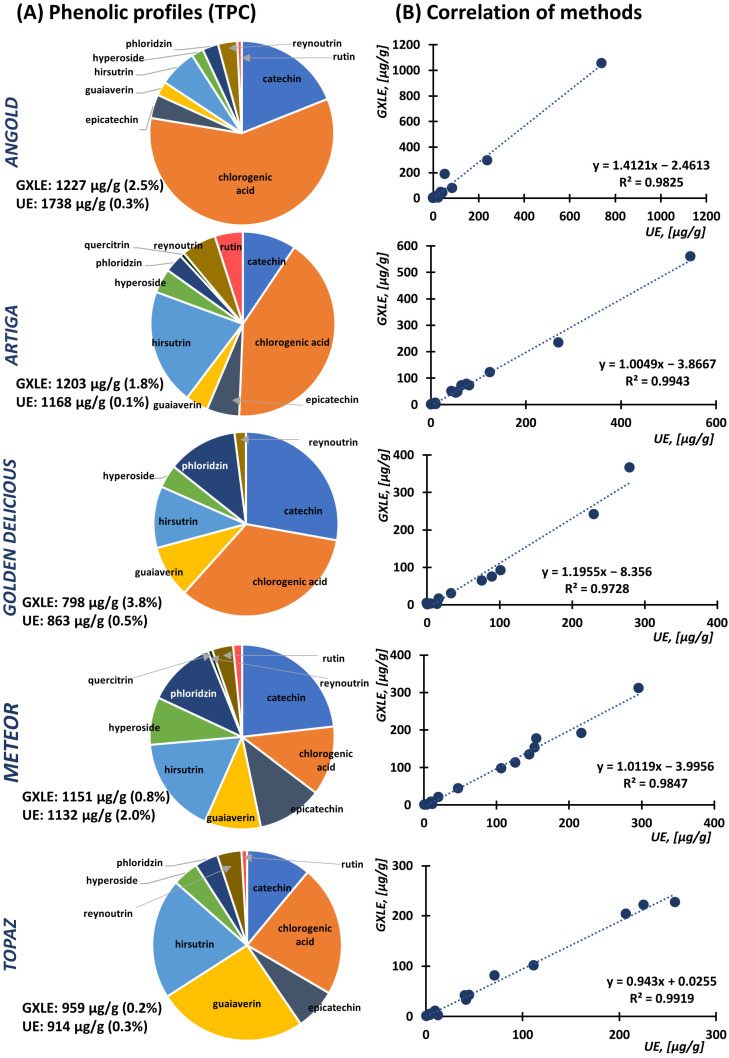
(**A**) Comparison of phenolic profiles for individual apple varieties. The sum (100%) corresponds to GXLE extracted amount. The RSDs are shown in the brackets for both TPC obtained by GXLE and UE. (**B**) Correlation of extracted amounts for individual compounds from each variety using GXLE and UE.

**Figure 3 foods-12-00893-f003:**
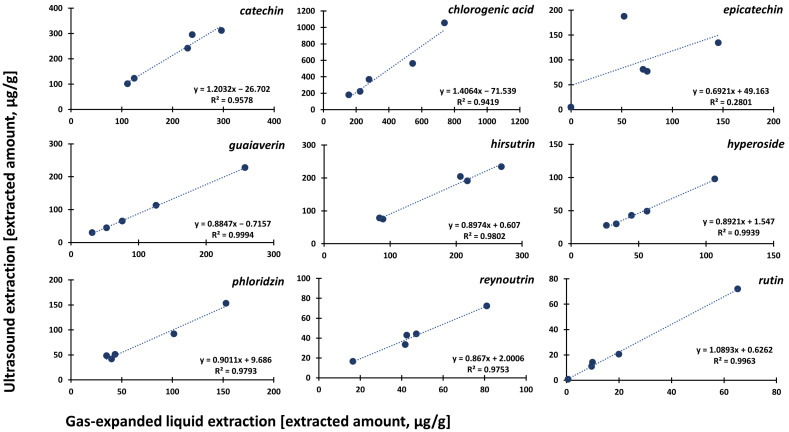
Correlation of extracted amounts for individual compounds from dried apples using GXLE and UE upon application of linear regression.

**Figure 4 foods-12-00893-f004:**
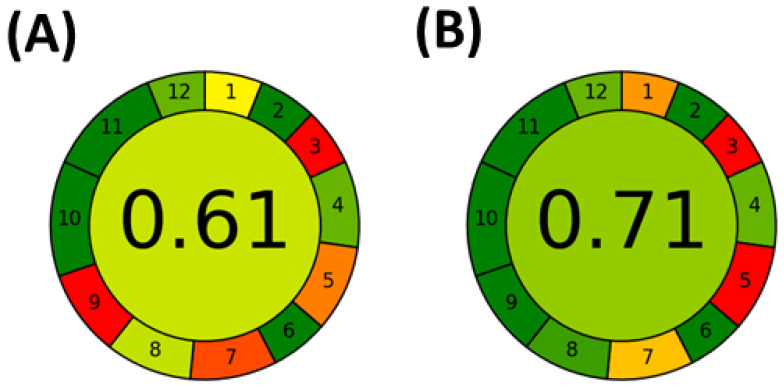
Evaluation of greenness of (**A**) GXLE a (**B**) UE plotted using AGREE software [32].

**Table 1 foods-12-00893-t001:** Set of experiments for SFE/GXLE optimization using Box-Behnken design.

Experiment	Run Order	CO_2_ [vol. %]	H_2_O [vol. %]	T [°C]	P [bar]	Extracted Amount [µg/g] *
N1	8	10	5	55	200	932
N2	14	70	5	55	200	427
N3	5	10	20	55	200	1495
N4	19	70	20	55	200	603
N5	3	40	12.5	30	100	987
N6	7	40	12.5	80	100	1254
N7	10	40	12.5	30	300	1085
N8	4	40	12.5	80	300	1306
N9	9	10	12.5	55	100	598
N10	20	70	12.5	55	100	734
N11	13	10	12.5	55	300	1531
N12	22	70	12.5	55	300	429
N13	2	40	5	30	200	270
N14	25	40	20	30	200	711
N15	26	40	5	80	200	1413
N16	12	40	20	80	200	1600
N17	23	10	12.5	30	200	608
N18	17	70	12.5	30	200	626
N19	27	10	12.5	80	200	887
N20	15	70	12.5	80	200	629
N21	16	40	5	55	100	771
N22	18	40	20	55	100	1437
N23	24	40	5	55	300	828
N24	6	40	20	55	300	635
N25	21	40	12.5	55	200	1143
N26	11	40	12.5	55	200	1349
N27	1	40	12.5	55	200	1302

* Extracted quantities summarized as a total phenolic content, including all tested bioactive compounds.

**Table 2 foods-12-00893-t002:** Set of experiments for UE optimization using D-optimal design.

Experiment	Run Order	EtOH in Water [vol. %]	T [°C]	Extracted Amount [µg/g] *
N1	9	0	30	1668
N2	3	100	30	616
N3	8	0	70	2011
N4	4	100	70	1366
N5	5	0	57	1885
N6	1	100	43	810
N7	7	100	57	928
N8	10	66.6	30	1747
N9	12	33.3	70	2237
N10	11	66.6	70	1759
N11	15	50	50	1836
N12	6	50	50	2096
N13	13	50	50	1855
N14	2	50	50	1883
N15	14	0	30	1650

* Extracted quantities summarized as a total phenolic content, including all tested bioactive compounds.

## Data Availability

Data is contained within the article or Appendix A.

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
