# Peer review of "Green Solvents in the Extraction of Bioactive Compounds from Dried Apple Cultivars"

_foods, 2023, doi:10.3390/foods12040893_

Round 1

Reviewer 1 Report

Comment to Author:

1.       Title: Modify your title in a scientific way.

2.       Abstract: Could be more concise and simpler.

3.       Abstract: Use of abbreviations are unsystematic and unclear that impairs good readability.

4.       Introduction: Must be less ambiguous and more precise about the topics that will be specifically addressed.

5.       30: Reference is missing

6.       Introduction. There is a lack of references here, most in deep research to introduce the work must be done.

7.       Line number 47-50. There is no reference for this statement, please check these paper and cite:

High intensity ultrasound treatment to produce and preserve the quality of fresh-cut kiwifruit. Journal of Food Processing and Preservation, 46(5), e16542.

Advances in green processing of seed oils using ultrasound-assisted extraction: A review. Journal of Food Processing and Preservation, 44(10), e14740.

Sonication, a Potential Technique for Extraction of Phytoconstituents: A systematic review. Processes, 9(8), 1406.

Ultrasonication as an emerging technology for processing of animal derived foods: A focus on in vitro protein digestibility. Trends in Food Science and Technology, 124, 309-322.

8.       Why did you select those extraction methods? randomly?

9.       In introduction add scientific name of all varieties of apple.

10.   Line number 106. Add city and country for blender.

11.   Mention overall profile of apple phenolic compounds which going to be extracted

12.   Reference studies are missing in results and discussion portion. How you compare your present results with previous studies?

13.   Results and discussion should be improved.

14.   Conclusions must be rewritten.

Author Response

Reviewer 1

Point 1.       Title: Modify your title in a scientific way.

Response 1: We have modified the title to more correspond to out aims.

“Green solvents in extraction of bioactive compounds from dried apple cultivars”

Point 2.       Abstract: Could be more concise and simpler.

Response 2: We have revised nomenclature in abstract, but we still try to keep all details that can help readers to find out quickly what is our article about.

Point 3.       Abstract: Use of abbreviations are unsystematic and unclear that impairs good readability.

Response 3: Abbreviations were unified in terms of extractant mixtures to correspond with SFE, GXLE and UE principles. The other abbreviations - UHPLC-DAD, EtOH - were omitted in Abstract and explained in the Materials and methods part.

Point 4.       Introduction: Must be less ambiguous and more precise about the topics that will be specifically addressed.

Response 4: Introduction part was revised, and our main aims were specified in more precise way.

Point 5.       30: Reference is missing

Response 5: The reference was added into this general sentence focused on sample matrix and its relation to extraction types.

Point 6.       Introduction. There is a lack of references here, most in deep research to introduce the work must be done.

Response 6: We have revised references, mention more up-to-date articles and more precisely specified out main aims and their relationship with the previously published works.

Point 7.       Line number 47-50. There is no reference for this statement, please check these paper and cite:

High intensity ultrasound treatment to produce and preserve the quality of fresh-cut kiwifruit. Journal of Food Processing and Preservation, 46(5), e16542.

Advances in green processing of seed oils using ultrasound-assisted extraction: A review. Journal of Food Processing and Preservation, 44(10), e14740.

Sonication, a Potential Technique for Extraction of Phytoconstituents: A systematic review. Processes, 9(8), 1406.

Ultrasonication as an emerging technology for processing of animal derived foods: A focus on in vitro protein digestibility. Trends in Food Science and Technology, 124, 309-322.

Response 7: Thank you for the listed references. Concerning main aims of our work we have selected two of them related to our work – see references [11-12].

Point 8.       Why did you select those extraction methods? randomly?

Response 8: Ultrasound extraction was selected as the most often applied, very simple and accessible for lab use. On the other hand, SFE for extraction of polar substances was chosen as green alternative to routine extractions while mainly GXLE principle is used for decreasing polarity of carbon dioxide in mixtures with co-solvents. This way green solvent are extractants in both approaches and their comparison is feasible. The manuscript text was modified (lines 91-93).

Point 9.       In introduction add scientific name of all varieties of apple.

Response 9: The scientific name – Malus domestica – was added to the Introduction text (line 28).

Point 10.   Line number 106. Add city and country for blender.

Response 10: The specification of the used blender was completed (line 135).

Point 11.   Mention overall profile of apple phenolic compounds which going to be extracted.

Response 11: Apple phenolic profiles are well-known and differ among apple cultivars while main phenolic substances were selected in our standard mixture. Then, for the respective cultivar the obtained phenolic profiles were compared with our previous study while differences in phenolics content caused by humidity and sunshine in apples harvested in the respective year were observed. But the main phenolics in the respective cultivars were the same. ‘Angold’ was characterized by high content of chlorogenic acid while in other cultivar other compounds were in the substantially higher amount – catechin, guaiaverin, and hirsutrin. Despite the phenolic profiles were included just to prove that extractions are comparable for different apple cultivars, brief comparison with the previous studies was discussed in Results and the text was revised following this comment.

Point 12.   Reference studies are missing in results and discussion portion. How you compare your present results with previous studies?

Response 12: Comparison in terms of extraction procedures and mainly applied extractants based on non-toxic solvents was already mentioned in the Discussion and Introduction part was modified according to the reviewer comment (lines 97-99). Comparison of the obtained phenolic profiles was carried out just with our previous study (text was revised in the Discussion part) as phenolic profiles were added just to prove applicability of the developed extraction methods (lines 431-436).

Point 13.   Results and discussion should be improved.

Response 13: The Results and discussion part was modified and revised following the comments of all reviewers and we hope that corresponds to the main aim of the article.

Point 14.   Conclusions must be rewritten.

Response 14: Conclusion part was revised and rewritten.

Reviewer 2 Report

Interesting work, very well conducted in methodological terms, however the proposed title “Green approaches for extraction of bioactive compounds from dried apple cultivars”, is not in line with the objectives “The aim of our study was development of UE and SFE using design of experiment approach enabling multivariate optimization of selected parameters to assess the effect of individual parameters as well as their possible interactions”.

Numerous similar studies have already been carried out with other plant matrices (optimization, extractive methods), mainly when it comes to using “residues” for the extraction of bioactive compounds, obtaining a value-added product from a residue. The proposal to optimize 2 different extraction processes of 5 apple fruit cultivars (not apple residue) is valid, however, perhaps the focus of the work would have to be on the development of a methodology using the apple as an experimental matrix.

In fact “Green approaches” in the methodology was a well justified term, but it would have been even better if it were applied to a by-product; in such case, it would be possible to unite “Green” meant for the extraction methods to the study object (residue).

I believe that unifying Title, objectives and conclusions in terms of developing a “new methodology” would be the most appropriate.

Some methodological aspects have to be better explained.

Was any type of treatment performed to prevent the loss of bioactive compounds (antioxidants) from the apple before the extraction processes? Apples, from the moment they are cut/processed, begin their oxidation process (polyphenoloxidase, etc.). Did you take this situation into account? If so, do explain.

2.3 Dried apple samples

Line 105: The apple slices were grinded using a powerful kitchen blender (Sencor Super Blender SBU 7730BK) to obtain powder with homogeneously distributed parts of apple peel and pulp.

A puree, not powder, was obtained.

Line 107: A stock of homogenized dried apple samples was stored in sealed containers in the dark at 4 °C and used for the optimization of extraction methods. Individual samples of the same cultivars were prepared using the identical protocol and extracted by applying optimized extraction conditions.

How and under what conditions were the apples/puree dried? This needs to be explained.

How were the variables chosen (temperatures, pressure, etc.)? Did you have any reference methodology, which one?

Author Response

Reviewer 2

Interesting work, very well conducted in methodological terms, however the proposed title “Green approaches for extraction of bioactive compounds from dried apple cultivars”, is not in line with the objectives “The aim of our study was development of UE and SFE using design of experiment approach enabling multivariate optimization of selected parameters to assess the effect of individual parameters as well as their possible interactions”.

Numerous similar studies have already been carried out with other plant matrices (optimization, extractive methods), mainly when it comes to using “residues” for the extraction of bioactive compounds, obtaining a value-added product from a residue. The proposal to optimize 2 different extraction processes of 5 apple fruit cultivars (not apple residue) is valid, however, perhaps the focus of the work would have to be on the development of a methodology using the apple as an experimental matrix.

In fact “Green approaches” in the methodology was a well justified term, but it would have been even better if it were applied to a by-product; in such case, it would be possible to unite “Green” meant for the extraction methods to the study object (residue).

Point 1: I believe that unifying Title, objectives and conclusions in terms of developing a “new methodology” would be the most appropriate.

Response 1: The manuscript title, objectives (lines 88-101 in track-changes revision), and conclusions (lines 470-786) were revised according to this comment.

Point 2: Some methodological aspects have to be better explained.

Was any type of treatment performed to prevent the loss of bioactive compounds (antioxidants) from the apple before the extraction processes? Apples, from the moment they are cut/processed, begin their oxidation process (polyphenoloxidase, etc.). Did you take this situation into account? If so, do explain.

Response 2: Indeed, the loss of bioactive compounds from samples can occur prior the extraction. For this reason, we worked with dried apple slices obtained from apple varieties received from (i) Research and Breeding Institute of Pomology, Holovousy, Czech Republic, and (ii) the local supermarket. The drying protocol used in the first case was included in the manuscript, lines 129-132. Unfortunately, we cannot know the procedure used for the preparation of the commercial products. After the apple slices were grinded, they were stored in a closed amber glass flasks in a darkness at 4 °C. To diminish the chance of a possible degradation, the extraction was carried out as soon as possible after the preparation of the powder from dry apple slices.

Point 3: 2.3 Dried apple samples, Line 105: The apple slices were grinded using a powerful kitchen blender (Sencor Super Blender SBU 7730BK) to obtain powder with homogeneously distributed parts of apple peel and pulp. A puree, not powder, was obtained.

Response 3: We grinded the already dried apple samples as described in 2.3. Thus, after their grinding, we obtained the powder.

Point 4: Line 107: A stock of homogenized dried apple samples was stored in sealed containers in the dark at 4 °C and used for the optimization of extraction methods. Individual samples of the same cultivars were prepared using the identical protocol and extracted by applying optimized extraction conditions.

How and under what conditions were the apples/puree dried? This needs to be explained.

Response 4: The drying protocol is now described in the manuscript, lines 129-132.

“They were cut to the slices and immediately dried in Steba ED fruit dryer with five drying plates and electronic temperature control. The first step of pre-drying lasted for about 1 h at 50 °C defined by the water content in the raw material. Then, the main drying step was accomplished at 60 °C for 7 h.”

The selected conditions are in agreement with our Utility model No. 33395, 2019 (https://www.vsuo.cz/images/patenty/jablen_produkt_suen.pdf, available in Czech language), where the effect of temperature in drying of apples and stability of selected phenolics was detailed.

Point 5: How were the variables chosen (temperatures, pressure, etc.)? Did you have any reference methodology, which one?

Response 5: We considered (i) solvent composition, (ii) extraction temperature, and (iii) pressure for SFE, that are the main parameters affecting the solvent physicochemical properties and, therefore, the solubility and extracted amounts of selected compounds. As we discussed in the manuscript, lines 265-272, the solvent composition was selected based on the physicochemical properties of analytes, mainly taking the polarity into account. Moreover, we wanted to emphasize the “greenness” of the method. Thus, ethanol and water, considered polar green solvents were selected. Temperature and pressure were selected based on the instrument limitations and CO2 critical point (temperature range: laboratory temperature to 90°C, max. pressure limit 400 bar). The tested temperature ranged from 30 to 80 °C, that is again in accordance with our Utility model No. 33395, 2019. The pressure did not exceed 300 bar to avoid a total over-pressurizing of the system. Indeed, during equilibration the system typically reaches the value higher than the set one and actual pressure can attack the pressure limit of the system before achieving the initial conditions of the extraction.

For UE, the temperature and solvent composition were selected again with stability, solubility, and physicochemical properties of analytes in mind. As mentioned in manuscript, lines 343-346, frequency as a main parameter could not be optimized since it was fixed in the instrument used.

Reviewer 3 Report

In this study, supercritical CO2 extraction and ultrasonic extraction are used to optimize the extraction conditions for phenolic compounds from five varieties of apples.

The type and ratio of entrainer, temperature, and pressure were optimized for supercritical CO2 extraction, while the temperature and the mixing ratio of the solvents, ethanol and water, were optimized for ultrasonic extraction.

Before judging the content, first of all, there are too many difficult-to-read passages throughout the manuscript.

You should outline around line 76 what evaluation of greenness is. The situation in line 385, which can finally be understood by looking at the references cited, is due to lack of explanation. At least around line 76, it is necessary to explain what aspects of evaluation of greenness are valid methods, the concept and outline of the evaluation of greenness method and its novelty.

It should clearly state sCO2, not SFE; SFE could be water or low-boiling-point organic solvents.

Chapter 2.

The order of the sections is inappropriate and very difficult to read.

It would be easier to understand if the order was "samples and chemicals => experimental methods =>analysis".

2.3 Dried apple samples

2.1 Chemicals and reagents

2.6 Extraction methods

2.4 Analysis of the active compounds using UHPLC-DAD

2.2 Preparation of the standard solutions

2.5 Design of experiments

2.6.1 Supercritical fluid extraction

Please describe the SFE system in detail. This is not a general-purpose system like analytical instruments, and information is only available on the web. If this manufacturer discontinues production of this model number, there will be no information about the construction of the system.

For example, we need the shape of the column, probably a cylinder, but we also need the inner diameter and length.

Please write what Dynamic extraction mode is. I think this term is specific to this instrument. If you mean semi-batch extraction of columns, rather than batch extraction, please explain it clearly.

Please also write the volume of the dried and milled sample (0.5 g). I do not know the contact time.

I don't know what co-solvent is in line 155. co-solvent is not mentioned anywhere in chapter 2.1 - line 155. Reading the Abstract, I think CO2/EtOH is co-solvent, but it is not shown in the main text, which is very confusing.

Tables 1 and 2

It is very confusing that the results are written in the experimental method.

I think this should be shown at the beginning of chapter 3.

I think that the order of the sections in chapter 2 became absurd and very difficult to understand because of the unreasonable attempt to show the results in chapter 2.6.

Do the numbers in Experiment and Run order make sense in the explanation that follows? Am I correct in understanding that N15 is connected to Table S2? If so, please explain it clearly in the text.

2.6.2 Ultrasound extraction

Please write the volume and output of the sonication bath. These are important information for reproduction of the experiment by other researchers.

Line 427-428.

Table S2. Optimized SFE conditions suggested by Optimizer for TPC in dried apples

Table S3. Optimized SFE conditions suggested by Optimizer for TPC in dried apples

If Table S3 is Probability of failure is 0, the caption must be changed to make it clear.

Author Response

Reviewer 3

In this study, supercritical CO2 extraction and ultrasonic extraction are used to optimize the extraction conditions for phenolic compounds from five varieties of apples.

The type and ratio of entrainer, temperature, and pressure were optimized for supercritical CO2 extraction, while the temperature and the mixing ratio of the solvents, ethanol and water, were optimized for ultrasonic extraction.

Before judging the content, first of all, there are too many difficult-to-read passages throughout the manuscript.

Point 1: You should outline around line 76 what evaluation of greenness is. The situation in line 385, which can finally be understood by looking at the references cited, is due to lack of explanation. At least around line 76, it is necessary to explain what aspects of evaluation of greenness are valid methods, the concept and outline of the evaluation of greenness method and its novelty.

Response 1: The greenness of the method is an established term that has been recently applied in evaluation of extraction procedures. Thus, we did not include the detailed information at “Introduction section”. We do not agree that the text in the discussion section about the greenness evaluation is unclear, as we provide detailed explanation, including evaluation protocols in Supplementary material, of greenness evaluation. The “greenness” evaluation was included as a part of the work and for the possibilities of other evaluation, the readers are referred to very detailed references.

Nevertheless, main aims were more specified in the terms of the “greenness” evaluation (lines 97-99).

Point 2: It should clearly state sCO2, not SFE; SFE could be water or low-boiling-point organic solvents.

Response 2: In manuscript, it is clearly said, that SFE mainly uses supercritical CO2 (sCO2), as you can see in Introduction (lines 65-70), main aims (lines 88-92), methodology, where used methods are specified (lines 265-267), as well Results and discussion. In SFE, similar to SFC, sCO2 is considered as a solvent of a choice. The SFC/SFE community suggested and uses the term “supercritical fluid chromatography, i.e., extraction” for techniques, using sCO2 with or without of small addition of co-solvent, despite the fact we talk not only about sCO2, but also about subcritical CO2+co-solvent fluid. The used fluid has gradually changing properties, doesn’t matter the co-solvent (alcohols, acetonitrile, etc.) amounts varying from 0 to 40 % (v/v). The reviewer is kindly referred to: Berger, J Chrom A, 2015, 1421, 171 (https://doi.org/10.1016/j.chroma.2015.07.062) and Lesellier, West, J Chrom A, 2015, 1382, 2 (dx.doi.org/10.1016/j.chroma.2014.12.083) for very detailed information.

Moreover, sCO2 can be used in mixtures with organic solvents with or without water addition in various ratios ranging from super/subcritical fluids to gas expanded liquids. Thus, when necessary, we changed the terminology and the term gas-expanded liquid extraction (GXLE) was used throughout the manuscript text.

Neat water and organic solvents are not typically used under supercritical conditions, as they are not easily achieved under laboratory conditions and/or are very corrosive. For more details, please see reference [13] in our manuscript.

Chapter 2.

Point 3: The order of the sections is inappropriate and very difficult to read.

It would be easier to understand if the order was "samples and chemicals => experimental methods =>analysis".

2.3 Dried apple samples

2.1 Chemicals and reagents

2.6 Extraction methods

2.4 Analysis of the active compounds using UHPLC-DAD

2.2 Preparation of the standard solutions

2.5 Design of experiments

Response 3: We do not agree with the reviewer. We checked the order of the subchapters carefully and we prefer to keep them as they are in accordance with our previously published works and publication practice.

2.6.1 Supercritical fluid extraction

Point 4: Please describe the SFE system in detail. This is not a general-purpose system like analytical instruments, and information is only available on the web. If this manufacturer discontinues production of this model number, there will be no information about the construction of the system.

For example, we need the shape of the column, probably a cylinder, but we also need the inner diameter and length.

Response 4: The instrument is already described, including the original name, producer, and all included modules that is typical for all published papers. See lines 184-188. Moreover, we included pressure and temperature limitations of the system in the text of manuscript. As it is commercially available system, all additional information of the interest can be found on manufacturer’s websites.

Typically, the extraction vessel is specified by the volume. Nevertheless, we asked the producer for the more detailed information about the extraction vessel size, and we added to the manuscript (line 194).

Point 5: Please write what Dynamic extraction mode is. I think this term is specific to this instrument. If you mean semi-batch extraction of columns, rather than batch extraction, please explain it clearly.

Response 5: Dynamic mode is an established term in the theory of extraction. The methods can be carried out in a static mode, where the defined volume of solvent is added to the sample without any application of flow rate, i.e. maceration, incubation, etc., while dynamic mode is considered when the flow rate is applied, i.e. SFE, PLE, etc. Please, se following references: DOI: 10.1016/s0021-9673(03)01106-3, 10.1016/s0021-9673(03)01106-3, or https://doi.org/10.1016/j.chroma.2020.461770.

Point 6: Please also write the volume of the dried and milled sample (0.5 g). I do not know the contact time.

Response 6: We did not measure a sample volume as it was weighted, we do not consider this value as relevant. If the reviewer means the extraction time, where the sample is in the contact with a solvent, the extractions took 10 min during the method optimization, and 30 min under final conditions in GXLE, and 10 min in UE.

Point 7: I don't know what co-solvent is in line 155. co-solvent is not mentioned anywhere in chapter 2.1 - line 155. Reading the Abstract, I think CO2/EtOH is co-solvent, but it is not shown in the main text, which is very confusing.

Response 7: Indeed, originally the information in line 155 describes the pump of the system, so we did not include the co-solvent type. The term “co-solvent” is fully explained in the introduction: “polar co-solvent miscible with sCO2 has to be added from negligible amounts to high levels at which the supercritical state is not achieved”, see lines 68-71. Co-solvent type used in our study is discussed in Lines 201-205: “(ii) water addition to EtOH as a co-solvent mixture in the range 5 – 20 %,”

Tables 1 and 2

Point 8: It is very confusing that the results are written in the experimental method.

I think this should be shown at the beginning of chapter 3.

I think that the order of the sections in chapter 2 became absurd and very difficult to understand because of the unreasonable attempt to show the results in chapter 2.6.

Response 8: Tables 1 and 2 were moved to the Results part (Chapter 3).

Point 9: Do the numbers in Experiment and Run order make sense in the explanation that follows? Am I correct in understanding that N15 is connected to Table S2? If so, please explain it clearly in the text.

Response 9: Yes, numbers of experiments are listed from 1 to 15, or 27 based on the table. The run order means when the sample was carried out. For example, table 1, experiment N27 was carried out as the first one, followed by N13, etc. This Run order randomizes the samples order to avoid the statistical independence among the results.

We carefully checked whole manuscript, and there is no N15 in the text. Reviewer probably meant the experiment O15, referred to the Table S2 in Included in Supplementary material, that is already described in the manuscript, line 303.

2.6.2 Ultrasound extraction

Point 10: Please write the volume and output of the sonication bath. These are important information for reproduction of the experiment by other researchers.

Response 10: The specification of the sonication bath, type and supplier, are described in the manuscript (DU-32 (Argo Lab, Italy), line 223) and the other details were added to the text – ultrasonic power 120 kW, frequency 40 kHz and volume 3.2 L.

Line 427-428.

Point 11: Table S2. Optimized SFE conditions suggested by Optimizer for TPC in dried apples

Table S3. Optimized SFE conditions suggested by Optimizer for TPC in dried apples

If Table S3 is Probability of failure is 0, the caption must be changed to make it clear.

Response 11: The title of Table S3 was corrected according to the comment, in fact Table S3 deals with UE conditions.

Reviewer 4 Report

Two extractions were reported in this work; however, I have some doubts about the description of the called “supercritical fluid extraction”. Different compositions for carbon dioxide - ethanol - water mixtures were applied in the extraction of phenolic compounds from dried apples:

This kind of extraction must be supported by a phase envelope to ensure supercritical conditions of the CO2-EtOH-H2O mixture. Some mixtures have low CO2 composition and conditions could indicate a compressed liquid phase and consequently the manuscript writing must be changed.

Introduction: Some references are not updated (6,7,14-19). The “new trends” must be supported by current research.

The literature survey did not reflect the manuscript purpose. What is the goal of the present manuscript? Is it contrasting extraction methods, maximization of extraction, identification of phenolic compounds?

2.6. Report the particle distribution analysis for the studied samples.

2.6.1. How was the CO2 and ethanol pumped? Has each one its own pump module? Explain. The same comment for water feeding.

Tables and figures. Units are missing or are incomplete (% mass or volume or mole) in some figures.

Author Response

Reviewer 4

Point 1: Two extractions were reported in this work; however, I have some doubts about the description of the called “supercritical fluid extraction”. Different compositions for carbon dioxide - ethanol - water mixtures were applied in the extraction of phenolic compounds from dried apples: This kind of extraction must be supported by a phase envelope to ensure supercritical conditions of the CO2-EtOH-H2O mixture. Some mixtures have low CO2 composition and conditions could indicate a compressed liquid phase and consequently the manuscript writing must be changed.

Response 1: We agree with the reviewer that supercritical conditions were probably not achieved under all tested conditions. Nevertheless, several reports were published that focused on carbon dioxide as a part of the extraction solvent., They included not only SFE, but also gas-expanded liquid extraction (GXLE) where CO2 was not major part of the extraction solvent. Following the comment of this reviewer, we changed the terminology across the entire manuscript. Moreover, we plotted all conditions tested in Design of experiments and final optimized conditions in the ternary phase diagram (J. S. Lim and Y. Y. Lee, J. Supercrit. Fluids, 1994, 7, 219–230.) to evaluate whether we could obtain single phase extraction solvent.

Point 2: Introduction: Some references are not updated (6,7,14-19). The “new trends” must be supported by current research.

Response 2: We updated the references according to the reviewer’s advice and the Introduction was revised. The added references focused on the application of green solvents in both extraction procedures we optimized, i.e., UE and SFE/GXLE. New references are [7-12] and [18-20].

Point 3: The literature survey did not reflect the manuscript purpose. What is the goal of the present manuscript? Is it contrasting extraction methods, maximization of extraction, identification of phenolic compounds?

Response 3: As mentioned at the end of introduction, we aimed at the development of UE and SFE/GXLE as suitable approaches for the extraction of phenolic compounds from apples. We wanted to evaluate effect of the individual parameters, the method repeatability, and applicability to different cultivars with the emphasis on the maximisation of amount of the extracted compounds. Moreover, we included the method greenness to the final comparison since the importance of this parameter has been increasing in recent years. Thus, literature survey was extended to discover, how often the methods we tested were used in extraction of similar compounds in food matrix to reveal the trends in this field.

Point 4: 2.6. Report the particle distribution analysis for the studied samples.

Response 4: We did not carry out the particle distribution analysis since it was not important for our research. The sample is not available anymore to report it. The grinded powders of dried apples were sieved and only particles with a size fraction of 0.5-2.5 mm were used in the extraction experiments. We added this information in the text, lines 135-136 in track-changes revision.

Point 5: 2.6.1. How was the CO2 and ethanol pumped? Has each one its own pump module? Explain. The same comment for water feeding.

Response 5: Both solvents were pumped by fluid delivery module for pumping CO2 and the co-solvent as already mentioned in manuscript, lines 184-187. This module comprises high pressure pump for CO2 that is cooled, and another pump for a co-solvent, as well as fluid delivery module for pumping CO2 and the co-solvent.

Point 6: Tables and figures. Units are missing or are incomplete (% mass or volume or mole) in some figures.

Response 6: All figures and tables were carefully checked, and the units completed.

Round 2

Reviewer 1 Report

Great work. 

Reviewer 3 Report

The authors fully explained and complied with the necessary modifications.
As for the order of the experimental methods, I still disagree with the author's answer and consider it very difficult to read in light of my experience. But it is not a fatal problem, so it is acceptable.

Reviewer 4 Report

The manuscript has been corrected